# Development of strategic social information seeking: Implications for cumulative culture

**Kirsten H. Blakey**[¤a]*, **Eva Rafetseder**, **Mark Atkinson**[¤b], **Elizabeth Renner**[¤c], **Fía Cowan-Forsythe**, **Shivani J. Sati**, **Christine A. Caldwell**

Faculty of Natural Sciences, Psychology, University of Stirling, Stirling, United Kingdom

¤a Current address: Faculty of Arts and Humanities, Philosophy, University of Stirling, Stirling, United Kingdom
¤b Current address: School of Management and School of Psychology & Neuroscience, University of St Andrews, St Andrews, United Kingdom
¤c Current address: Department of Psychology, Durham University, Durham, United Kingdom
* kirstenhblakey@gmail.com

**Data Availability Statement:** All data and analysis files are available in an OSF repository which can be accessed here: https://osf.io/ctse7/.

## Abstract

Human learners are rarely the passive recipients of valuable social information. Rather, learners usually have to actively seek out information from a variety of potential others to determine who is in a position to provide useful information. Yet, the majority of developmental social learning paradigms do not address participants' ability to seek out information for themselves. To investigate age-related changes in children's ability to seek out appropriate social information, 3- to 8-year-olds ($N = 218$) were presented with a task requiring them to identify which of four possible demonstrators could provide critical information for unlocking a box. Appropriate information seeking improved significantly with age. The particularly high performance of 7- and 8-year-olds was consistent with the expectation that older children's increased metacognitive understanding would allow them to identify appropriate information sources. Appropriate social information seeking may have been overlooked as a significant cognitive challenge involved in fully benefiting from others' knowledge, potentially influencing understanding of the phylogenetic distribution of cumulative culture.

## Introduction

Seeking out relevant information from appropriate social sources is ubiquitous in human adults. Human adults may demonstrate key differences in the way they seek, attend to, and use social information compared to children and non-human animals (henceforth animals). This propensity for identifying and gathering relevant social information has been proposed as one of a suite of cognitive mechanisms that may be required for distinctively human cumulative culture [1–3]. By 'distinctively human' cumulative culture we are referring to the accumulation of beneficial modifications to cultural traits over successive generations of learners, which results in increased functionality or efficiency [4–6]. Its rarity in animals and apparent importance in accounting for human evolutionary success has prompted interest regarding the emergence and development of cognitive mechanisms in human children that support

**Funding:** KHB: PhD studentship funded by the Division of Psychology, University of Stirling. CAC: 648841 RATCHETCOG ERC-2014-CoG European Research Council https://erc.europa.eu/ The funders had no role in study design, data collection and analysis, decision to publish, or preparation of the manuscript.

**Competing interests:** The authors have declared that no competing interests exist.

cumulative culture [5,7,8]. It may be the lack, limited scope, or inflexibility of mechanisms such as information seeking that impedes development of human-like cumulative culture in animals despite them showing evidence of culture [9–11] and social learning abilities [12,13]. The current study aimed to address the gap in our understanding regarding age-related changes in children's ability to seek out and use appropriate social information. Documenting the developmental trajectory of this capacity could provide insights into the cognitive demands involved and whether it is likely to be observed in animals.

In a bid to understand the discontinuity between humans' and animals' capacity for cumulative culture a number of sociocognitive mechanisms have been considered [4,7,14,15]. Some of these proposals focus on when and how social information is used. Social learning is generally considered to be adaptive when it is present within a population. However, models have shown that under most conditions it is the ability to flexibly switch from social to individual learning, when social learning proves unsatisfactory, that increases adaptability in a population [16–20]. Selective or flexible rules that influence individuals' use of social information are known as social learning strategies (SLSs). These refer to heuristic biases or rules that dictate when, what, and from whom social information should be acquired [21,22], helping to filter out less useful aspects of available social information. The selective nature of SLSs such as 'copy older individuals' or 'copy the majority' makes them generally more effective than learning indiscriminately from others or individual learning. However, extensive evidence of SLSs in both young children and animals [22–26] suggests that selective social learning cannot account for the marked differences we see between humans and animals with regard to the capacity for cumulative culture.

One proposal that attempts to explain the distinctiveness of human cumulative culture outlines a dual-process view of social learning. This view, put forward by Heyes [27], suggests that there are distinct categories of SLSs each based on specific types of decision rules. According to the first half of this account, distinctively human cumulative culture could be attributed to the use of explicitly metacognitive SLSs [27,28]. These are rules that are consciously represented and (for verbal individuals) reportable, that reflect learners' explicit awareness and understanding of the social learning strategies they are employing. Learners who employ an explicitly metacognitive SLS have been proposed to depend on theory of mind and advanced metacognitive capacities that enable human adults to flexibly identify, select, or disregard social information across varied contexts [2,27–29]. As such, these strategies afford learners the capacity to represent what information is required and to reason about where, or from whom, that information can be obtained. General SLSs such as 'copy older individuals' do not necessarily take into account the knowledge or experience of other individuals with regard to a particular task. By contrast, we propose that explicitly metacognitive SLSs afford learners the ability to flexibly reflect on others' potential to provide valuable information by taking into consideration their knowledge, experience, or access to information in relation to a particular task. Thus, learning can be targeted towards the most appropriate source of information (e.g., when learning to use information technology one should learn from 'digital natives' [30]).

The second part of Heyes's [27] dual-process account of social learning proposes that the majority of the behaviours that conform to SLSs in both humans and animals are based on decision rules that depend on general-purpose associative learning processes or biologically selected biases. These rules direct learning towards objects, agents, and events that are most likely to provide useful information across a broad range of contexts. However, unlike explicitly metacognitive SLSs, these *implicit SLSs* (as we will refer to them here) are not driven by a causal understanding of the potential value of social information. Although we refer to these strategies as implicit, we do not claim that learners employing them are necessarily devoid of insight regarding personal preferences that guide their social learning. It is likely that learners

sometimes explicitly represent strategies related to salient yet superficial cues without appreciating *why* their strategy is successful. However, we suggest that such strategies, while they may be explicitly represented, are not explicitly metacognitive due to the absence of a causal understanding of informants' potential to provide valuable information. Social learning biases in young children and animals are likely driven by such implicit, and relatively crude, heuristic decision rules [31]. Indeed, cases in which very young children [32–36] and animals [24,26,36,37] select knowledgeable others are likely the result of implicit SLSs (e.g., associative learning), rather than the reasoning-based strategies employed by adults. For example, model-based biases for older [25,32,38], higher-status [23,34], or even more reliable models [35] may be the result of repeated exposure to the successes of models with these characteristics, resulting in rule-like strategies.

While there is evidence of implicit SLSs, based on heuristic biases, in both young children and animals, in our view, there is as yet no solid evidence of explicitly metacognitive SLSs in either population. By contrast, adult humans are (with good reason) assumed to be able to use social information in an explicitly metacognitive manner (although this should not be taken to mean that they do not also use implicit SLSs). For example, it is routine for human adults to actively *seek out* models of social behaviour, using their understanding of others' knowledge, access to information, or intentions relative to their own to strategically select and use the most relevant social information [39,40]. A critical question, then, is how children's use of social information develops with age, particularly in relation to their explicit understanding of the value of information that can be provided by others, and how they might be able to benefit from it. Since we assume that this understanding is not present from infancy, but is certainly in place by adulthood, it follows that this transition must occur over the course of childhood. We may therefore be able to identify key stages during development when children begin to change how they respond to social information. This would provide insights into the cognitive capacities upon which such abilities depend, and therefore might also shed light on the reasons for the apparent absence of these abilities in animals.

Research into social learning has largely been restricted to investigating the circumstances under which social information is used, the efficacy of that use, and its role in cultural transmission [41–47]. Indeed, to date, developmental research into SLSs has focused on examining children's responses to task-relevant social information (usually an effective solution) which is provided in advance of an opportunity to solve the same task [44,48–51]. However, while cultural transmission necessitates using information acquired from a social source, human learners are rarely passive recipients of valuable social information. Here, we propose that *actively* seeking out valuable information when faced with a particular problem to solve is more analogous to real world social learning scenarios, compared with being *passively* provided with relevant information.

Therefore, in the current study we were particularly interested in determining when children develop the ability to seek out social information based on an understanding of its value. This led us to consider selective information seeking paradigms that already exist in the literature. Specifically, we looked at the selective trust paradigm. This paradigm is commonly used to examine children's preferences for information provided by models with conflicting social and/or epistemic characteristics, and is built on the premise that children learn from others' testimony (see [52] for a full review).

These studies typically employ a 'conflicting sources paradigm' in which children first observe two informants who differ on social (e.g., gender or accent) and/or epistemic (e.g., accuracy or reliability) characteristics [35]. Following a familiarisation phase, children are faced with an unfamiliar scenario, for example, naming an unfamiliar object. In some tasks, children are required to select one of two potential informants to 'seek' information from (i.e.,

'ask' questions) and/or required to make their selection following the informants' claims about the name of the object (i.e., 'endorse' questions). The model selected by the child is considered to be the model whose claim they trust. With regards to the influence of social characteristics on selective trust, evidence suggests that in the absence of epistemic differences children ask and endorse informants who have positive social characteristics [53]. These selections are influenced by both the models' and the learners' own characteristics (e.g., age or gender [54]). In particular, social characteristics that signal a model's similarity to the learner (i.e., ingroup membership) are consistently favoured by young children (e.g., preference for informant with a native accent [55]; preference for informant of the same gender [56]). Model-based biases such as these are unlikely to require explicit cognitive reasoning. Consistent with the selective preferences in children's proclivity to copy and the SLS 'copy successful individuals', they are likely the result of implicit biases that promote learning from sources that are most likely to provide useful information across the broadest range of contexts. The selective trust literature reports that children as young as 3 years old are sensitive to informants' social and epistemic characteristics [33,35,55,57]. Recent meta-analyses examined the relation between children's age and their selective trust decisions [53]. Results indicated that children asked and endorsed more knowledgeable (accurate/reliable) informants when they differed on only epistemic characteristics. Specifically, 4-year-olds were more likely than 3-year-olds to endorse knowledgeable informants. When informants differed on epistemic and social characteristics simultaneously, 4- to 6-year-olds were more likely to endorse informants who were knowledgeable but had a negative social characteristic while 3-year-olds appeared to weigh both characteristics equally. Thus, from 4-years children appear to place greater value on epistemic characteristics.

Whilst these preferences may be linked to a developing explicit awareness of the potential value of social information, the design of these studies precludes this conclusion. The paradigm depends on participants being exposed to information about the accuracy or reliability of the two informants in the familiarisation phase, in order to establish the respective epistemic characteristics (e.g., knowledgeable/ignorant) of the conflicting sources. Thus, the literature on selective trust, like much of the literature on SLSs, depends on children making choices between models on the basis of characteristics for which they are likely to have a prior history of associations or a pre-existing bias (whether established as part of the experimental procedure, e.g., reputation for accuracy, or from the child's own life experience, e.g., age [58]). The results can therefore be likewise attributed to implicit biases. However, the apparent transition to favouring epistemic characteristics (such as prior accuracy) over social ones (such as familiarity [59]) is perhaps suggestive of a developing insight into the value of others as sources of social information.

While it is useful to know how children use social information and who children prefer to learn from, we argue that this is not sufficient to determine the cognitive mechanisms that children are employing during social learning. Children's ability and proclivity to seek out *relevant* social information has not yet been adequately addressed. That is, nothing in the SLSs or selective trust literature has examined children's ability to select valuable social information on the basis of its relevance for solving a specific problem.

Explicitly metacognitive SLSs are proposed to be experience-dependent and learned through social interaction, so we would expect them to emerge relatively late in development [27]. Therefore, to help explain why cumulative culture appears to be restricted to humans we can look at whether age-related changes in children's ability to seek out relevant social information coincide with advances in cognitive development. If we find particular ages at which children make significant advances in their appropriate social information seeking and these occur at a similar age to the development of particular cognitive capacities, then these capacities might be necessary prerequisites. Similarly to Heyes [27], Baldwin and Moses [1] proposed

that motivations for initiating an appropriate search for social information likely rest on advanced metacognitive capacities. They emphasised that to seek social information effectively, the seeker should have awareness of what information is required and from whom it can be obtained. While behavioural tests of implicit metacognitive ability suggest that infants [60], young children [61], and animals [62,63] react to the state of ignorance, they do so without necessarily recognising a metacognitive awareness of that state. However, we know that children's cognitive capacities continue to progress well beyond those of animals. In particular, as their cognition advances, children develop abilities such as evaluating their own knowledge state [64,65], understanding others' mental states [66,67], and recognition that perceptual access to information facilitates knowledge formation [68–70]. Such abilities are thought to be cognitively demanding; thus, their requisite nature may preclude younger children's (and by logical extension animals') ability to seek out appropriate social information. Therefore, if we can identify when children develop the capacity to use explicitly metacognitive over implicit SLSs we may be able to use this to predict the likelihood of the capacity being available to animals. That is, if we find that this ability develops late in childhood then this would be consistent with the hypothesis that it is dependent on cognitive capacities that are not available to animals, and could help to explain the distinctiveness of human cumulative culture.

The aim of the current study was to investigate when children develop the ability to seek out social information using explicitly metacognitive SLSs. To explore the development of children's appropriate information seeking we designed a task in which 3- to 8-year-old children were faced with a problem for which they could not use prior experience or knowledge. Rather, to solve the problem children had to reason about the information needed and who had the potential to provide that information. In certain circumstances, explicitly metacognitive SLSs dictate that social cues should be ignored, at least when cues are available which are more directly related to task relevant experience. This was the case for the task in the current study which required children to choose which of four possible demonstrators could provide critical information for unlocking a box. Specifically, children's choice between social demonstrations was related to characteristics of the box each of the demonstrators had access to (relative to the box they themselves were faced with), rather than personal characteristics of the demonstrator. Attending to the social demonstration was critical to children's successful use of the information. Thus, although the critical characteristics children had to attend to were not social, they functioned to determine what social information (in the form of the demonstrator's attempt) would provide the details they needed to solve the problem. It should also be noted that while choices made by matching one's own box with one of the demonstration options would appear to show appropriate information seeking, making such a choice without appreciating why it is useful is unlikely to result in successful use of the information. Understanding why a particular demonstration is useful involves some recognition of the value of the social information as it requires the child to understand that the demonstrator's performance (rather than just the box) will be informative. The demonstrator's success (or failure) in unlocking the box, was the key detail children required to unlock their own box.

Appropriate information seeking should be based on a reasoned understanding of the value, to oneself, of the social information. Heyes [27] argued that this kind of cognitive reasoning may not be available in young children or animals. In the absence of the ability to reason about the potential value of social information provided by others, we expected children to rely on less cognitively demanding implicit SLSs. In the current study we specifically looked at whether children's information seeking might instead be influenced by heuristic model-based biases for superficial demonstrator characteristics, such as age or gender. Thus, the task was designed such that the use of model-based biases in the absence of cognitive reasoning would be inappropriate, leading to imperfect information seeking.

Overall, we expected to find an age-related transition from the use of heuristic model-based biases (implicit SLSs) to reasoning-based choices driven by the value of the information (explicitly metacognitive SLSs).

Specifically, we predicted that appropriate information seeking would improve with age due to our anticipation that advances in children's explicit metacognition would enable them to identify appropriate sources of information based on an understanding of what information they required. We expected that younger children would struggle to employ explicitly meta-cognitive SLSs and instead rely on less cognitively demanding heuristic biases (implicit SLSs) related to superficial demonstrator characteristics such as age or gender. Any age-related changes in children's appropriate information seeking could indicate use of different SLSs when approaching the task. Thus, we examined the developmental trajectory of appropriate social information seeking to provide insight into the emergence of cognitive reasoning as a mechanism required for distinctively human cumulative culture. Finding evidence of reasoning-based choices in older, but not younger, children would support the proposal [27] that explicitly metacognitive SLSs are experience-dependent, developing relatively late in children.

Examining whether children use social information appropriately (following appropriate information seeking) could further help to distinguish between learners with some level of metacognitive understanding, and those reliant on implicit rules for what, when, and whom to copy. That is, understanding the relevance of the acquired information would be expected to also result in more appropriate *use* of that information. By investigating social information use in conjunction with appropriate information seeking we can really begin to expose the cognitive mechanisms that underlie these processes. How children use particular types of information provided by others can tell us much about the nature of their social learning processes–most importantly whether they might be using reasoning-based strategies or relying upon adaptive heuristic biases. For instance, children who have sought out information based on an understanding of its value are more likely to be able to use it appropriately than children who selected it on the basis of an implicit SLS. In the current study, copying was not always the correct response, and success was sometimes dependent on making a different choice to the one made by the demonstrator. As with appropriate information seeking, we expected successful use of the social information to improve with age.

## Method

### Participants

The final participant sample comprised 218 children aged three to eight years (112 females; Mean age = 71.1 months, *SD* = 20.4 months, range = 37 to 106 months). The sample size was balanced across age groups: 3 years (*n* = 36), 4 years (*n* = 40), 5 years (*n* = 37), 6 years (*n* = 36), 7 years (*n* = 35), 8 years (*n* = 34). The sample includes children recruited from a school in Scotland (*n* = 118) and visitors recruited at Edinburgh Zoo (*n* = 100). The nationality of the sample, as identified by parents or legal guardians, was predominantly British. An additional eight children were tested but later excluded from analyses due to researcher or technical errors (*n* = 4), missing data (*n* = 2), and task interference (*n* = 2). This study was approved by the University of Stirling General University Ethics Panel (GUEP555), and informed written consent was provided by each participant's parent or legal guardian.

### Materials

The task apparatus is presented in Fig 1. Four identical wooden boxes (16cm$^3$) were locked using child friendly plastic padlocks (Alphabet Learning Locks, Lakeshore, Carson, CA, USA). Each target padlock was a different colour (red, blue, orange, purple) with a different cartoon

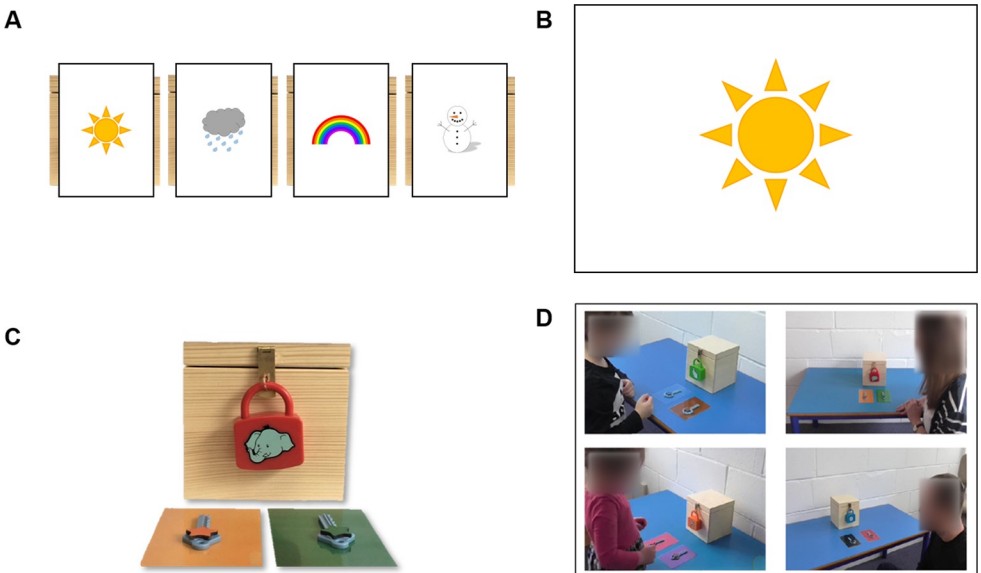

**Fig 1. Example of the experimental set-up for one of the four trials.** (A) Trial indicator cards placed in front of boxes. (B) Trial indicator image displayed on screen. (C) Box locked with a distinctive padlock, and two possible keys (one correct). (D) Still images presented on the screen representing available demonstration videos (the faces of the demonstrators have been blurred for publication), in this example the target video is on the top right.

image (elephant, queen, pig, sun) on the front. Additional padlocks of different colours and with different cartoon images were used for non-target social demonstration videos. One locked box was presented in each trial along with two plastic keys. Only one key could unlock the padlock. A coloured shape (star, rectangle, circle, oval) was fixed to each key: within a trial the same shape was fixed to both keys each in a different colour. So that the colours of the keys were salient in the demonstration videos, each key was placed on a rectangular laminated card of the same colour as the shape on each key. Each box contained a coloured rubber duck. Retrieved ducks were placed on a laminated image of a pond. A Lenovo Yoga 520 touchscreen laptop running PsychoPy v1.84.2 [71] in tablet mode was used to present visual stimuli and videos.

## Social demonstration videos

A total of 20 demonstration videos were created, including two possible target videos (successful and unsuccessful) and three non-target videos for each of the four trials. Each video depicted a demonstrator faced with a similar set-up to the participant, selecting one of two keys to try to unlock a padlock to open a box. Within each trial the age and gender of the four potential demonstrators was different (adult male, adult female, child male, child female; child demonstrators were all between 3 and 4 years) and each demonstrator had a different combination of padlock and keys from one another. In each trial a single target video showed a combination of padlock and keys which matched the participant's combination of padlock and keys, and three non-target videos in which none of the apparatus matched that of the participant. Thus, the target video would provide useful information regarding which key would unlock the padlock, and the non-target videos would provide irrelevant information. The age and gender of the target demonstrator was different in each trial and the order of the trials was randomised between participants. The arrangement of the target and non-target demonstrators on the screen was also randomly assigned between trials and participants. A successful

demonstration showed the selection of the key that unlocked the padlock and the box being opened, while an unsuccessful demonstration showed the selection of the key that did not unlock the padlock and the box remaining closed. Target videos were randomly allocated as successful or unsuccessful in each trial so that a participant would see a maximum of two videos of each demonstration type if they selected all four target videos. The three non-target videos in each trial comprised both successful and unsuccessful demonstrations.

## Procedure

Participants took part individually in a single session that lasted approximately 10 minutes. Two experimenters were present, experimenter 1 (E1) provided instructions and presented materials to participants, while a second experimenter controlled the laptop and live coded participants' responses. In some sessions, a familiar adult was also present but was asked not to interact with the participant during the task. Participants received a sticker for taking part.

At the outset each of the four locked boxes were placed behind one of four trial indicator cards (sun, rain, rainbow, snow) that corresponded to the trial in which that box was to be attempted (Fig 1A). The participant touched the screen to begin the task, generating one of the trial indicator images (Fig 1B) and allowing E1 to retrieve the corresponding box and keys for that trial from behind the trial indicator card. The box was brought to the front of the testing table and the two keys were placed on the correspondingly coloured cards side by side (Fig 1C) between the box and the participant (keys were presented on the same sides as in the associated target demonstration video). E1 explained that one of the keys would unlock the padlock so the box could be opened and one of the keys would not unlock the padlock so the box would stay closed. Participants were told that they would watch a video before trying to open their own box. E1 explained that participants would *"see four pictures of other people who have boxes they want to unlock"* and that they could *"choose one of the pictures to see a video of that person trying to open their box"*. Participants were instructed to *"choose a video that will help you choose which key to try and open your box with"*. The laptop screen was turned away during the instructions to reduce distraction. When it was turned to face the participant, four still images from the beginning of each demonstration video were presented on the screen (one target and three non-target; Fig 1D). Participants were asked *"Which video are you going to choose?"* to prompt them to select one of the videos to watch. Once an image had been selected, the corresponding demonstration video played immediately. When the video ended the screen turned white and was turned away; participants were told, *"Now it's your turn to try and unlock your box"* and asked, *"Which key are you going to choose?"* If participants chose the target key and successfully opened the box, they retrieved the rubber duck. If participants chose the non-target key the box remained locked. The same procedure was repeated for each of the four trials. Following completion of all trials E1 asked *"How were you deciding which videos to watch?"* and *"How were you deciding which keys to use?"* to probe whether children were able to explicitly and reasonably justify their choices.

## Statistical analysis

The analyses were performed in R [72], with generalised linear mixed effects analyses (GLMMs) performed using the *lme4* package [73] with logit regression. P-values < .05 were accepted as statistically significant. The binary dependent variables in the analyses were: appropriate information seeking (whether on each trial children selected the target video) and appropriate information use (whether on each trial children, who had selected and watched the target demonstration video, selected the target key). Where specified as fixed effects the following variables were sum coded: participant's gender (female as −1, male as 1), participant-

target demonstrator gender congruence (incongruent as −1, congruent as 1), target demonstrator age (child as −1, adult as 1) and demonstration outcome (unsuccessful as −1, successful as 1). Age was centred and scaled to measure thousands of days, and trial number was centred. See supplementary information for model outputs with standardised coefficients for age and trial number. Random effects were included to ensure that models accounted for variability between trial types and variability among individuals due to the repeated-measures design [74]. Therefore, the random effects structure for each model aimed to include by-participant random slopes for all fixed effects and keep random effects structures 'maximal' where possible [75]. Where the 'maximal' model resulted in non-convergent or singular fit models, random slopes were removed, followed by random intercepts where necessary, until a convergent, non-singular model was obtained. Post hoc analyses were carried out using estimated marginal means using the *emmeans* package [76]. Post hoc results are given on the log odds ratio scale.

## Results

The two key aims of this study were to examine whether children were able to seek out and select the target demonstration video and, if successful, whether they were able to use the social information in the demonstration to select the target key to unlock the padlock. We were also keen to explore whether the non-target demonstration video selections that children were making were driven by model-based biases for particular demonstrator characteristics. Finally, we examined children's responses to post-test questions probing their explicit reasoning.

### Information seeking

Appropriate information seeking was measured by the proportion of trials in which children selected the target video. This required children to reason about the information they needed to solve a problem (unlock their box) and which of the demonstrators could provide that information (which of the demonstrators were facing the same problem). If children had appropriately reasoned about the information they required, they should have selected the video which would give them information about the box they were faced with, and as such, disregarded demonstrator characteristics such as age and gender. Overall children selected the target demonstration video in 65% of 872 trials.

A GLMM was built for information seeking success with fixed effects of age, gender, participant-target demonstrator gender congruence, target demonstrator's age (the interactions between these variables), trial number (and its interaction with age), and a by-participant random slope for target demonstrator's age. The model was significantly better than the null equivalent ($\chi^2(17) = 123.6$, $p < .001$). A significant main effect of age ($b = 2.92$, $SE = 0.46$, $z = 6.32$, $p < .001$) indicated that children's appropriate information seeking improved with age (Fig 2), with 7- and 8-year-old children successfully selecting the target video in 90% and 91% of trials, respectively. Binomial tests revealed that appropriate information seeking was significantly above what would be expected by chance (25%) in each age group ($p < .001$). A significant main effect of trial number ($b = 0.32$, $SE = 0.11$, $z = 2.93$, $p = .0034$) indicated that children showed better information seeking in later trials. There was also a significant two-way interaction between age and trial number ($b = 0.37$, $SE = 0.18$, $z = 2.04$, $p = .041$). To clarify this interaction, we performed a post hoc analysis using *emmeans*. This indicated that the effect of trial (looking specifically at the difference between the first and last trial) was present in older (upper quartile; $b = 1.50$, $SE = 0.49$, $z = 3.05$, $p = .0023$) but not younger children (lower quartile; $b = 0.36$, $SE = 0.35$, $z = 1.03$, $p = .303$).

A significant main effect of participant-target demonstrator gender congruence ($b = 0.38$, $SE = 0.12$, $z = 3.26$, $p = .0011$; Fig 3) suggested that children selected the target video

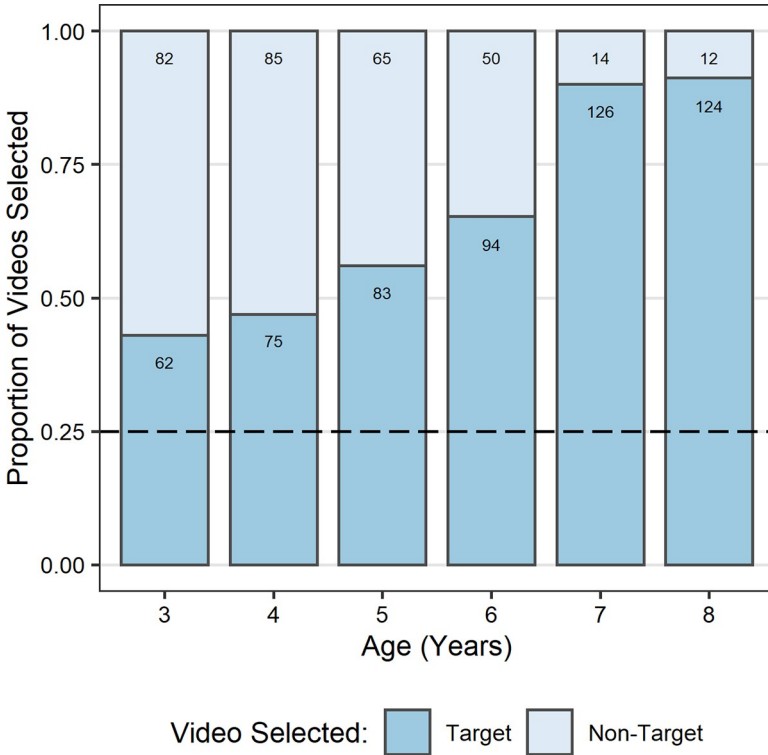

**Fig 2. Proportion of target and non-target demonstration video selections (N = 872) by age in years.** Target video selections equate to appropriate information seeking. All age groups selected the target video significantly above chance. Values presented on bars refer to the number of trials each proportion represents. Dashed line indicates chance.

significantly more often when the gender of the target demonstrator was congruent ($M = .69$, $SD = .46$) rather than incongruent ($M = .60$, $SD = .49$) with their own. The GLMM did not reveal any evidence of an overall effect of gender ($b = -0.16$, $SE = 0.20$, $z = -0.82$, $p = .412$) or the target demonstrator's age ($b = -0.10$, $SE = 0.20$, $z = -0.50$, $p = .618$) on appropriate information seeking. However, there was a significant two-way interaction between gender congruence and participant gender ($b = -0.23$, $SE = 0.12$, $z = -1.97$, $p = .049$). To clarify this interaction, we performed a post hoc analysis using *emmeans*. This indicated that the gender congruence effect was significant in female ($b = 1.22$, $SE = 0.35$, $z = 3.50$, $p = .0005$) but not male participants ($b = 0.29$, $SE = 0.31$, $z = 0.93$, $p = .350$).

A three-way interaction between gender congruence, target demonstrator age and participant gender was also found to be significant ($b = 0.33$, $SE = 0.12$, $z = 2.76$, $p = .006$). To clarify this interaction, we performed a post hoc analysis using *emmeans*. This indicated that the gender congruence effect found in female participants was restricted to cases in which the target demonstrator was a child ($b = 1.67$, $SE = 0.55$, $z = 3.04$, $p = .013$), while male participants showed a significant gender congruence effect when the target demonstrator was an adult ($b = 1.16$, $SE = 0.43$, $z = 2.70$, $p = .035$). For an overview of the distribution of demonstrator selections split by participant age and gender see Fig 4.

## Information use

Appropriate information use was measured by the proportion of trials in which children, who had selected and watched the target demonstration video, selected the target key. The

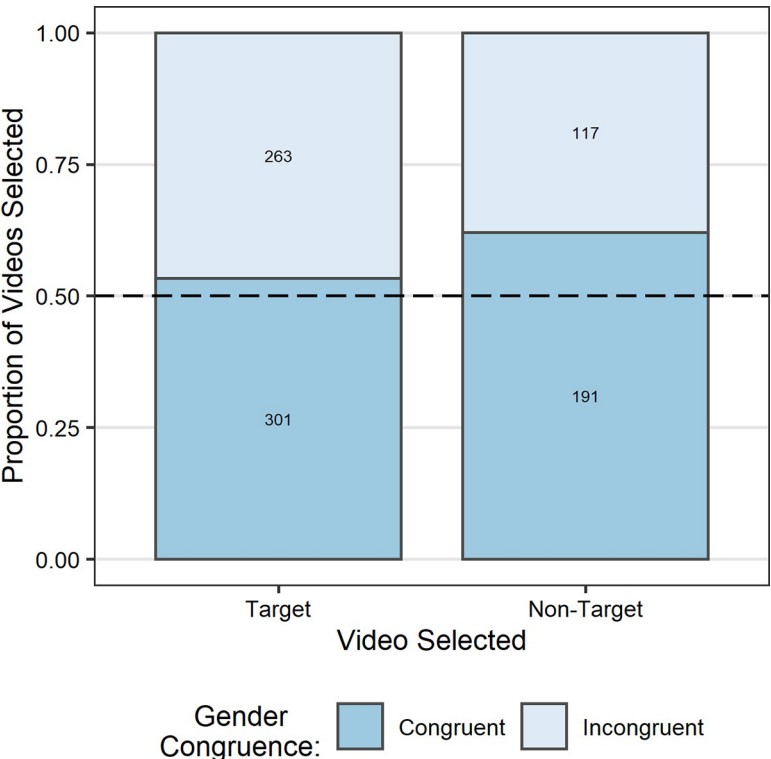

**Fig 3. Proportion of target and non-target demonstration video selections (*N* = 872) by participant-selected demonstrator gender congruence.** Target video selections equate to appropriate information seeking. Values presented on bars refer to the number of trials each proportion represents. Dashed line indicates chance.

appropriate response was dependent on the success of the demonstration. Viewing a successful demonstration should have encouraged a copying response (using either the colour or the position of the key), while viewing an unsuccessful demonstration should have encouraged avoidance of that response, and selection of the alternative. Overall children chose the target key in 83% of 564 trials (287 successful and 277 unsuccessful demonstrations).

A GLMM was built for appropriate information use with fixed effects of age, demonstration outcome, and the interaction between these variables, and random intercepts of participant gender, target demonstrator (adult male, adult female, child male, child female), and participant ID. The model was significantly better than the null equivalent ($\chi^2(3) = 72.5$, $p < .001$). A significant main effect of age was revealed ($b = 1.86$, $SE = 0.30$, $z = 6.26$, $p < .001$) indicating that appropriate information use improved with age (Fig 5). The model also identified a significant main effect of demonstration outcome ($b = 0.69$, $SE = 0.18$, $z = 3.89$, $p < .001$) indicating that children were more successful following successful demonstrations ($M = .88$, $SD = .33$), than unsuccessful demonstrations ($M = .78$, $SD = .42$). It is worth noting that appropriate information use was particularly high in 6-, 7-, and 8-year-old children following successful demonstrations, with children successfully copying the demonstrator's key choice in >95% of trials. Finally, there was a significant two-way interaction between age and demonstration outcome ($b = 0.52$, $SE = 0.25$, $z = 2.09$, $p = .037$). To clarify the direction of this interaction, we performed a post hoc analysis using *emmeans*. This indicated that the effect of demonstration success was slightly more pronounced in older children (upper quartile; $b = 1.88$, $SE = 0.53$, $z = 3.52$, $p = .0004$) than younger children (lower quartile; $b = 0.85$, $SE = 0.29$, $z = 2.98$, $p = .003$).

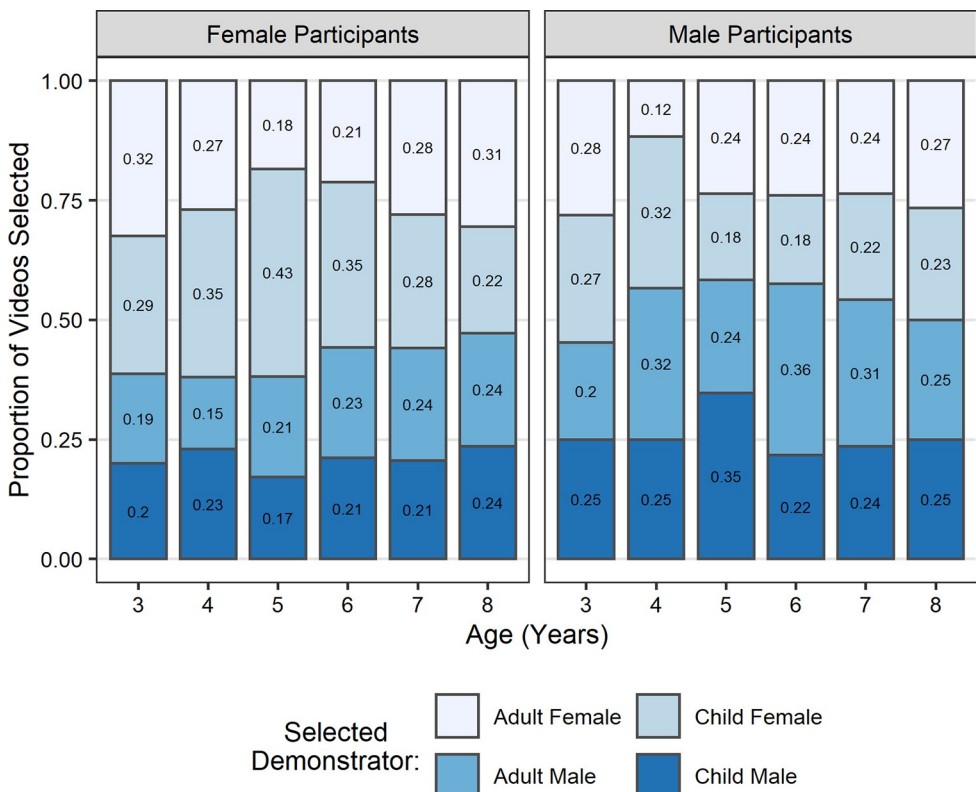

**Fig 4. Proportion of selections (*N* = 872) of each demonstrator by participant age and gender.** Optimal information seeking would have resulted in an equal distribution of selections of each demonstrator. Values presented on bars refer to proportion of selections in each section.

## Explicit verbal reasoning

We examined children's responses to explicit reasoning questions regarding their choices in both the information seeking and information use aspects of the task. Responses to each reasoning question were categorised into four levels: 0) no response; 1) non-reasoned response–responses that did not relate to the question asked (e.g., *'the good ones'/'I just used my brain'*), or comprised single words or gestures; 2) reasoned but incorrect responses–explanations that showed evidence of explicit reasoning but the motivations were incorrect or the answer was insufficient to determine full correct reasoning (e.g., *'picked one of the grown ups'/'because some are wrong and some are right'*); and 3) reasoned correct response–the explanation provided clear reasoned evidence of explicit task understanding (e.g., *'because they had the same lock'/'in the video, if they didn't get it I chose the other one, if they did I chose the same one'*). When asked to justify how they were selecting which videos to watch, 38% of children provided correct reasoned responses, and when asked to justify how they were selecting which keys to use, 23% gave correctly reasoned responses.

We investigated whether children's verbal reasoning was predicted by age or task performance by conducting two ordinal regressions (using the *ordinal* package in R [77]. For each reasoning question (information seeking and information use) we submitted children's responses to an ordinal regression with fixed effects of age, performance (total number of target video and target key selections, respectively), and the interaction between these variables. Both models indicated significant main effects of both age (information seeking: $b = 1.38$, $SE = 0.61$, $z = 2.24$, $p = .025$; information use: $b = 2.49$, $SE = 0.54$, $z = 4.61$, $p < .001$; Fig 6) and

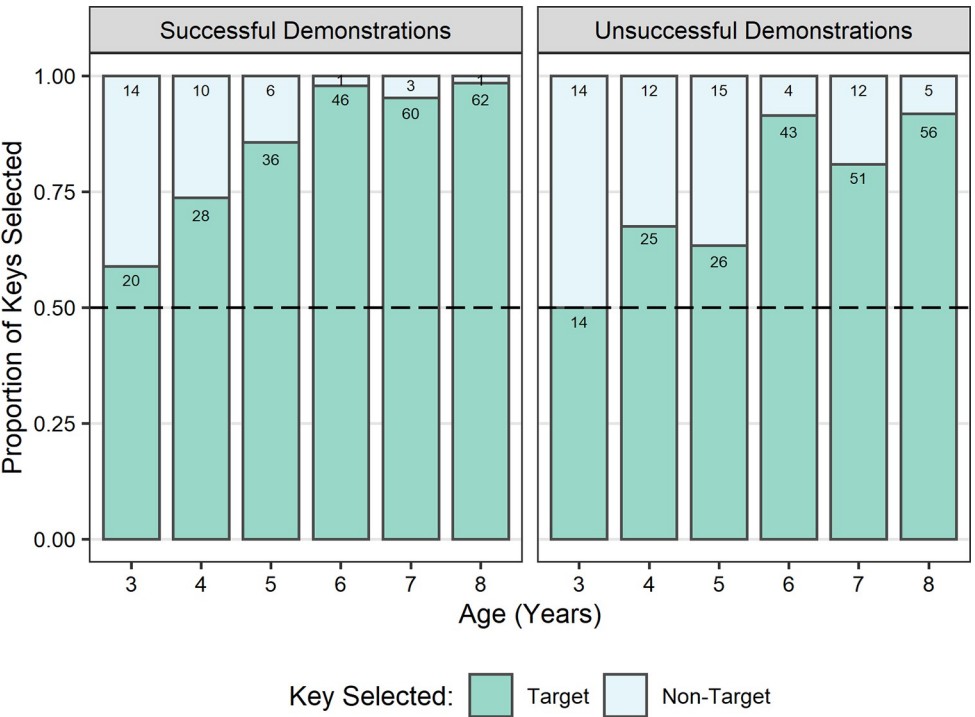

**Fig 5. Proportion of target and non-target key selections by age in years.** Includes only trials in which participants who watched the target demonstration video ($N = 564$). Target key selections equate to appropriate information use. After successful demonstrations, the appropriate response was to copy the demonstrator's key selection; after unsuccessful demonstrations, the appropriate response was to avoid copying the demonstrator's selection. Values presented on bars refer to the number of trials each proportion represents. Dashed line indicates chance.

performance (information seeking: $b = 0.59$, $SE = 0.12$, $z = 5.11$, $p < .001$; information use: $b = 0.39$, $SE = 0.11$, $z = 3.44$, $p < .001$; Fig 7). There was no evidence of any significant interactions between age and performance (information seeking: $b = 0.24$, $SE = 0.21$, $z = 1.18$, $p = .237$; information use: $b = -0.07$, $SE = 0.19$, $z = -0.35$, $p = .729$). These results suggest that in response to both reasoning questions older children and children who made more target selections provided better reasoned responses than younger children and children who made less target selections, respectively.

## Discussion

In this study we investigated the development of appropriate social information seeking in 3- to 8-year-olds. Specifically, we were interested in identifying when children develop the ability to seek out relevant information based on a reasoned understanding of who they should observe in order to solve a particular problem, based on the characteristics of the problems faced by the potential demonstrators, in relation to the problem faced by themselves (explicitly metacognitive SLSs). We were also interested in exploring the alternative strategies children rely on prior to this development, therefore we examined the possible influence of model-based biases (implicit SLSs), which (if the task had been understood) should have been disregarded.

We found that the proportion of trials in which children selected the target demonstration video increased with age. This corresponded to our prediction that older children would exhibit more appropriate social information seeking than younger children. Despite finding

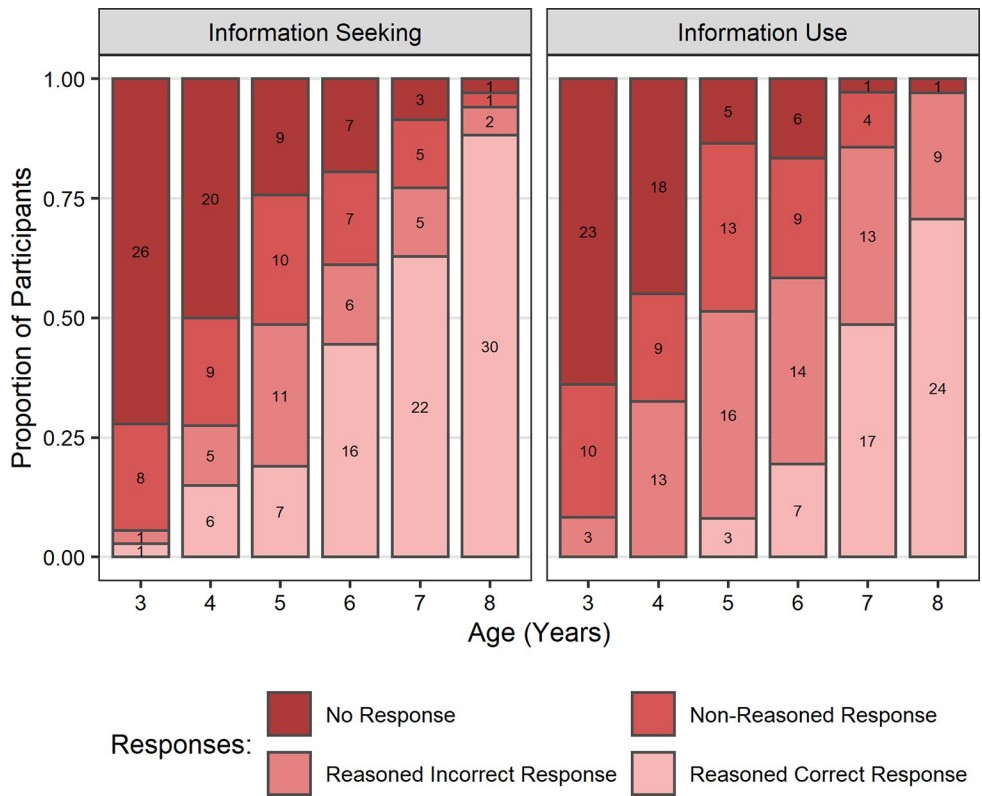

**Fig 6. Proportion of participants providing each category of explicit verbal reasoning responses by age in years and reasoning question.** All participants are included ($N = 218$) regardless of whether they sought out the correct information. Values presented on bars refer to the number of participants each proportion represents.

that all ages selected the target video significantly more often than would have been expected by chance, we saw a relatively high proportion of non-target video selections in the younger age groups. This suggests that older (7- to 8-years) and younger (3- to 6-years) children might have been approaching the task differently. The particularly high rate of appropriate information seeking we observed in 7- and 8-year-olds (close to ceiling) was consistent with our expectation that increased metacognitive understanding would allow them to recognise what information was required to solve the problem and to identify who, of the available demonstrators, could provide relevant information. Thus, we suggest that older children understood the potential value to themselves of the information that the target demonstrator could provide having employed an explicitly metacognitive SLS. By contrast, we postulate that younger children's target video selections were less likely to have been driven by reasoned understanding of the information they needed and the potential value of the target demonstration video. Therefore, when selecting a video, younger children might have instead relied on less cognitively demanding implicit SLSs such as model-based biases, or a bias for selecting models whose box matched their own box (without appreciating the value of the information). That is, although matching their own box and the one in the target video appears to indicate appropriate information seeking, if the selection was not based on an appreciation of the value of the information, then they would be expected to exhibit lower success in using that information appropriately.

The primary aim of this study was to establish when children develop the ability to seek out information based on a reasoned understanding of its value (explicitly metacognitive SLSs).

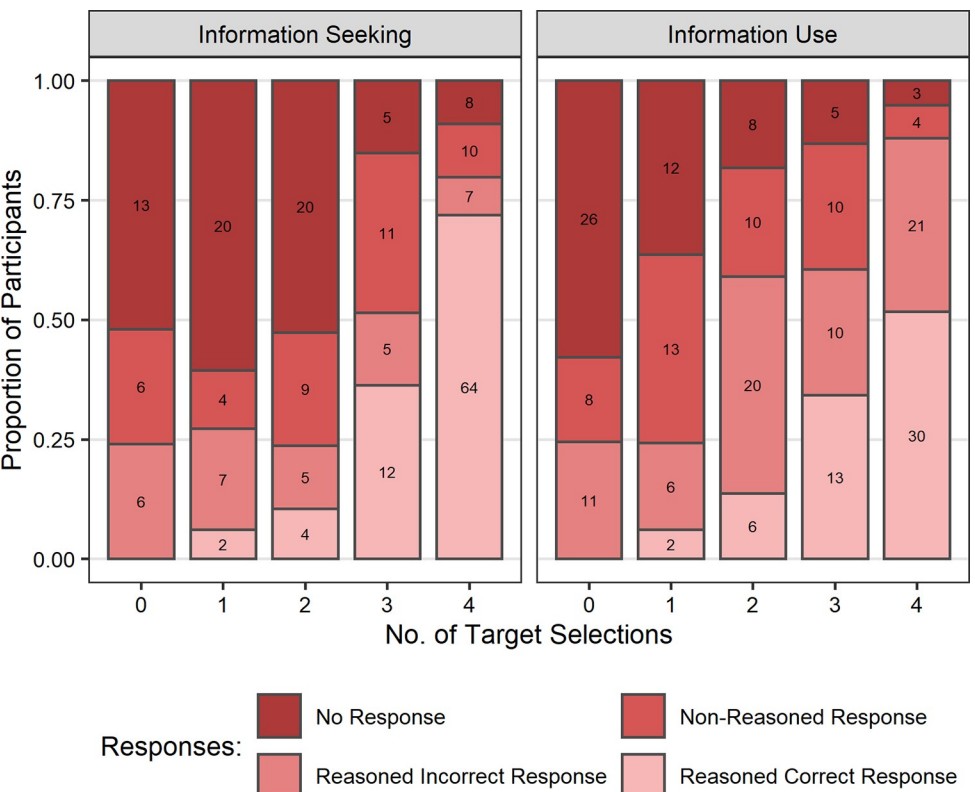

**Fig 7. Proportion of participants providing each category of explicit verbal reasoning responses by task performance and reasoning question.** Information Seeking: Responses by target video selections; Information Use: Responses by target key selection, target key selections were only possible after a target video selection. Values presented on bars refer to the number of participants each proportion represents.

However, we were also interested in what strategies children used to seek out information prior to the development of this ability. The design of this study made it possible to explore whether children might be relying on less cognitively demanding implicit SLSs such as heuristic model-based biases. If children understand the value of social information and its relevance to their goal, then they should be able to identify who can provide that information regardless of the characteristics (e.g., age or gender) of the available models. That is, appropriate information seeking based on an understanding of the value of the information should override model-based biases related to particular demonstrator characteristics. Therefore, in our analyses we assessed whether children's information seeking showed evidence of any biases towards either of the demonstrator characteristics we manipulated, namely age and gender. Results showed that overall children were more likely to select target demonstrators when they were the same gender as themselves, therefore showing a gender congruence effect. We found no overall bias related to the demonstrators' age. However, there were slight differences in particular demonstrator preferences between male and female participants; female participants showed the gender congruence effect when the target demonstrator was a child, while the effect was evident in male children when the target demonstrator was an adult. Therefore, it seems that even when it was apparent–from salient visual cues–which demonstrator could provide relevant information, some children were still driven by biases for superficial model characteristics (in this case primarily related to the demonstrators' gender) rather than the task-relevance of the information. These findings support the interpretation that in the absence of

the capacity for explicit reasoning about what information is needed and who can provide it, younger children were influenced by model-based biases. These biases are in line with the reports of preferences for same-gender models in selective trust paradigms [56]. This pattern of results suggests that children's target video selections were probably made as a result of model-based biases rather than reasoning about the value of the information that the demonstrator could provide.

As outlined above, we propose that the older children's high rate of appropriate information seeking was driven by reasoned understanding of the value of the information, whereas the younger children who have likely not yet developed the capacity to use these reasoning-based SLSs instead appeared to rely on model-based biases (in this case a preference for gender congruent demonstrators). The results of the current study highlight the importance of considering cognitive mechanisms involved in seeking out social information, as they reveal evidence of the different strategies that older and younger children use when approaching the task. Thus, we believe this provides evidence of a relatively late age-related transition from implicit SLSs to explicitly metacognitive SLSs, consistent with accounts proposed by Dunstone and Caldwell [28] and Heyes [27].

In line with our predictions, the proportion of trials in which children selected the target key (appropriate information use) increased with age. Rates of target key selections were found to be higher in older children following both successful and unsuccessful demonstrations, indicating that older children were more adept at flexibly using social information to inform their responses than younger children. We propose that older children knew what to do with the information they received in the demonstration because their appropriate information seeking had been based on an understanding of the information needed to solve their problem. Overall, these results provide further evidence for the interpretation that older children's social information seeking was based upon reasoned understanding of the value of the information (explicitly metacognitive SLSs). By contrast, younger children appeared to make less effective use of the information they acquired despite selecting the target video. We posit that this offers further evidence to support our interpretation that younger children's target video selections were less likely to have been driven by reasoned understanding of what information was required and the potential value of the target demonstration video. Rather, their selections might have been based on superficial but salient demonstrator characteristics driven by implicit SLSs (in this case, model-based biases for gender congruence).

Including a range of ages in our study allowed for investigation into age-related changes in children's use of SLSs, specifically when children develop the capacity to seek out information based on an understanding of its value. Previous research has tended to focus on a behaviour of interest in more limited age groups (e.g., only including 5-year-olds: [25,34]), therefore precluding investigation into developmental trajectories or age-related changes. Our results suggest that the transition from implicit SLSs to explicitly metacognitive SLSs begins at around 6 years. Compared to 7- and 8-year-olds, the proportion of information seeking errors (non-target video selections) made by 6-year-olds was still relatively high. However, for those 6-year-olds who did select the target video, their appropriate information use was at a level comparable to that of 7- and 8-year-olds for both demonstration types. This suggests that 6-year-olds who selected the target video were likely to have been motivated by a reasoned understanding of the relevance of the information or were at least able to recognise its value once the information had been received, indicative of explicitly metacognitive SLSs. Indeed, the non-target video selections in 6-year-olds (Fig 2) could be a reflection of the transition from implicit to metacognitive SLSs, with some children having not yet developed the capacity to use such strategies so relying on less cognitively demanding implicit SLSs. This interpretation would be consistent with reports of advances in metacognitive understanding in partial exposure tasks

[65] and tasks that require children to ascribe knowledge to others [68] from 6 years. Thus, we propose that cognitive advances in explicit metacognition are implicated in the ability to recognise the value of social information. These advances are related to the ability to reflect on one's own knowledge and, importantly, the ability to recognise when more information is needed to solve a particular problem, what that information is, and from where or whom it can be acquired. However, as we used age as a proxy for cognitive development and developmental trajectories of other cognitive capacities overlap with that of explicit metacognition, we cannot say for certain which cognitive developments might be implicated in the emergence of the capacity for explicitly metacognitive SLSs. Future studies should explore whether individual differences in the use of explicitly metacognitive SLSs are associated with differences on tasks directly measuring cognitive capacities such as mental state understanding and explicit metacognition.

Finally, by using the explicit reasoning questions, we examined children's own understanding of the strategies they were employing when approaching the task. The results not only show that explicit reasoning improves with age, but also that higher rated reasoning responses were associated with better performance in both the information seeking and information use aspects of the task. In both cases we found that older children gave more responses that reasonably justified the selections they had made. This supports our assumption that appropriate information seeking (observed in older children) is indicative of employment of an explicitly metacognitive strategy rather than reliance on implicit biases (observed in younger children) which are not related to an understanding of the potential value of information. As alluded to in the introduction, implicit SLSs may sometimes be explicitly represented. As such, information seeking responses that were rated as reasoned but incorrect provide an interesting insight into children's interpretations of their model selections. This type of response revealed explicitly represented justifications for particular model selections that were not linked to the potential to provide valuable information. Rather, these responses indicated use of strategies related to alternative superficial cues, that, for example, fit with the model-based biases we observed in the results (e.g., response from 5-year-old female: *'they were all girls'*). While these responses are explicitly represented, they are not explicitly metacognitive. Thus, these results are consistent with the account we have presented proposing that younger children relied on implicit SLSs that were not driven by reasoning about the value of the available information.

Overall, these findings suggest that children, like adults, can actively seek out relevant information in order to identify appropriate social models. Here we presented evidence that 3- to 8-year-old children's capacity to actively seek out appropriate social information from multiple potential models exceeds what we would expect to see if they were selecting a model at random. We believe that our results suggest that older and younger children might have been using different SLSs in the information seeking phase of the task. In older children (7- and 8-year-olds), social information seeking appears to have been driven by explicitly metacognitive SLSs pertaining to an understanding of the information required and the potential value of the information that could be provided. By contrast, younger children appeared to rely on less cognitively demanding heuristic model-based biases (implicit SLSs) due to having not yet developed the capacity to use reasoning-based strategies. The pattern of results observed in this study highlights the need for and the benefits of expanding developmental social learning paradigms to take account of information seeking as well as information use. By doing so we provided evidence of an age-related transition from use of implicit to explicitly metacognitive SLSs.

Finding that older, but not younger, children were making reasoned choices, based on the value to themselves of social information, supports the proposal [27] that explicitly metacognitive SLSs are experience-dependent and develop relatively late in children. The relatively late

emergence of the capacity to use explicitly metacognitive SLSs (though not necessarily the experience dependence) is also consistent with the idea that this may be a cognitive mechanism that is unique to humans. That is, the late development is an indication that these strategies depend on other cognitive developments which are also unique to humans. Thus, we propose that our findings are consistent with the interpretation that reasoned understanding of social information could account for distinctively human cumulative culture [1,27,28]. Real world instances of the evolution of cumulative culture are likely to involve learners acquiring social information that is very limited in its availability, almost by definition. Thus, it is not unreasonable to suggest that capacities for appropriate information seeking could determine whether or not a particular population exhibits cumulative culture. The late development of this skill, as identified in our findings, suggests that appropriate social information seeking may have been overlooked as a significant cognitive challenge to fully benefit from others' knowledge. It is possible therefore that this could provide part of the explanation for the phylogenetic distribution of cumulative culture.

## Supporting information

**S1 Table. GLMM outputs for standardised fixed effects.**
(DOCX)

## Acknowledgments

We are grateful to RZSS Edinburgh Zoo and West Lothian Council for the opportunity to recruit and test children. We also thank the teachers, parents, and children for their support and participation.

## Author Contributions

**Conceptualization:** Kirsten H. Blakey, Christine A. Caldwell.

**Formal analysis:** Kirsten H. Blakey, Mark Atkinson, Elizabeth Renner.

**Funding acquisition:** Christine A. Caldwell.

**Investigation:** Kirsten H. Blakey, Fía Cowan-Forsythe, Shivani J. Sati.

**Methodology:** Kirsten H. Blakey, Christine A. Caldwell.

**Project administration:** Kirsten H. Blakey, Christine A. Caldwell.

**Software:** Kirsten H. Blakey, Mark Atkinson.

**Supervision:** Eva Rafetseder, Christine A. Caldwell.

**Visualization:** Kirsten H. Blakey.

**Writing – original draft:** Kirsten H. Blakey.

**Writing – review & editing:** Kirsten H. Blakey, Eva Rafetseder, Mark Atkinson, Elizabeth Renner, Fía Cowan-Forsythe, Shivani J. Sati, Christine A. Caldwell.

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
