## [Decision Letter · Decision Letter 0]

5 May 2021

PONE-D-21-09287

Development of strategic social information seeking: Implications for cumulative culture

PLOS ONE

Dear Dr. Blakey,

I sent it to two experts in the area, and their comments appear below. As you can see, both reviewers were positively disposed towards the work, but have some comments that need to be addressed. As the reviews are clear and detailed I wont repeat their content here, but needless to say all points deserve attention.

I concur with the reviewers that your paper presents some very interesting and thought-provoking research that has the potential to make a contribution. I am also of the belief that you will be able to address the concerns raised and hence that your paper should become acceptable for publication pending suitable minor revision and modification of the article in light of the appended reviewer comments.

We look forward to receiving your revised manuscript.

Kind regards,

Mark Nielsen, Ph.D.

Academic Editor

PLOS ONE

Journal Requirements:

We note that Figure 1 includes an image of a participant in the study

Reviewers' comments:

Reviewer's Responses to Questions

**Comments to the Author**

1. Is the manuscript technically sound, and do the data support the conclusions?

Reviewer #1: Yes

Reviewer #2: Partly

2. Has the statistical analysis been performed appropriately and rigorously? 

Reviewer #1: Yes

Reviewer #2: Yes

3. Have the authors made all data underlying the findings in their manuscript fully available?

Reviewer #1: Yes

Reviewer #2: Yes

4. Is the manuscript presented in an intelligible fashion and written in standard English?

Reviewer #1: Yes

Reviewer #2: Yes

5. Review Comments to the Author

Reviewer #1: The manuscript investigates the development of strategic social information seeking in children aged 2-8. The authors report an experiment in which implicit SLSs would not be beneficial for the learner and, instead, metacognition would be necessary to perform well in the experiment. The authors consider that metacognition could explain why cumulative culture is present in humans but not in other animals. They provide a comprehensive literature review and rationale for their hypothesis. The experiment is well designed, and the statistical analysis are strong. In conclusion, this manuscript is a very valuable addition to the literature, and I recommend its publication. I only have a few minor comments and suggestions:

Line 74: Heyes’ dual process is introduced here. The first part of the process should be clearly stated here and Heyes should also be mentioned. She is mentioned later when explaining the second part of the dual process.

Lines 98-107: these lines are very interesting. I was wondering to what extent implicit SLSs are domain-general or domain-specific. Although the literature has found some evidence of copying of high prestige/older individuals over low prestige/younger individuals, this does not seem to be always the most adaptive strategy. For example, an old individual might prefer to learn IT from a younger individual than from an older one. Could implicit SLSs alone account for this domain-specificity or metacognition is necessary to fine-tune SLSs to each domain?

Lines 215-256. I understand you only have one hypothesis: there will be “an age-related transition an age-related transition from the use of heuristic mode-based biases to reasoning-based choices driven by the value of the information”. I think you could make this stand out more by stating the hypothesis in a separate paragraph.

Statistical analysis

Line 354: I think a definition of appropriate information seeking and appropriate information use is needed here. You do this later in the results section.

Lines 358-359: A sentence explaining why random effects where necessary for your experimental design would be beneficial for the reader.

Results

I would like to see some measure of effect size or to have a way to compare the strength of the different predictors in the regression models (e.g. standardized coefficients, using information criteria to compare the fit of different models taking in and out specific predictors). This could be reported in a table as supplementary materials.

I was wondering whether the section of information seeking errors is necessary. The results are the same as for information seeking: gender congruency is a predictor of information seeking, females prefer female models, but males do not show a preference for either males or females.

The symbol of chi squared does not look right: lines 436, 438, 443, 444, 447.

Line 500: ordinal (from ordinal package) needs italic.

499-512: Did you use random effects for the ordinal models? There is no mention of this here.

Figures 2, 3, and 5. I would like to see the numbers for the proportions for the different categories as this is shown in figures 4, 6, and 7.

Overall statement

I really like this manuscript. Congratulations to the authors for a very valuable contribution to the field of Social Learning.

Reviewer #2: This is a well-written paper on the development of “explicitly metacognitive” social learning strategies and their implications for the evolution of cumulative culture. The study was designed to investigate “when children develop the ability to seek out social information using explicitly metacognitive SLSs” (l. 215/216). The strong points of the study include its rather large sample size, the data availability, the use and clear description of appropriate data analysis methods, and the clear presentation and visualisation of the results. The introduction is long, but an interesting and informative read and demonstrates that the authors know the literature well and have grounded their research question well into theory. My comments relate mostly to the introduction (some concepts could be defined better), the study design (it is currently unclear why the chosen task investigates social information seeking rather than information seeking in general), and the discussion (some conclusions are not unequivocally supported by the data). I am especially unclear to what extent the task used in this study tests for the strategic use of social information – and this should be made much clearer. Many of my comments relate to this issue. However, I think these issues could be addressed in a minor revision. I added my comments point-by-point and chronologically below, with a few minor points at the end.

Introduction, l. 77/78: “Explicitly metacognitive SLSs are defined as consciously represented and reportable rules.” It would be great if this sentence could be specified to explain rules regarding what. Maybe an example could be added to help explain what explicitly metacognitive SLSs are.

Introduction, l. 80/81: “Such strategies involve the use of theory of mind and metacognitive processes”. Here, explicitly metacognitive SLSs are described as involving metacognitive processes, which is somewhat tautological. It remains unclear what is meant by metacognitive processes. I suggest replacing the wording “metacognitive processes” here with something more explanatory.

Introduction, l. 82-85: “As such, explicitly metacognitive SLSs afford learners the capacity to reason about what information is required and recognize the potential value to themselves of information that can be provided by others.” I find this description slightly contradictory – or at least not fully consistent – with the information provided further above in the introduction that SLSs can be described as rules, which assumingly help the individual e.g., to identify whom to learn from. Here, SLS are described as allowing the individual to reason about what information is required in the first place. This seems to me to be a different kind of demand, both in time (first I need to identify which information is needed, then I can use a rule that I can use to get that kind of information in my social environment) and in terms of the cognitive demands. It seems that there is a conflation between different demands: recognizing what information is required and using social learning strategies to help select the appropriate social information. Thus, I suggest describing in an even clearer way what exactly explicitly metacognitive SLS are and what they allow a learner to do (and also what it is not).

Introduction: It should also be clarified in what way explicitly metacognitive rules rely on language. The manuscript mentions that this type of SLSs consist of reportable rules – does this mean verbally reportable?

Introduction, l. 108-110: “While there is evidence of implicit SLSs, based on heuristic biases, in both young children and animals, in our view, there is as yet no solid evidence of explicitly metacognitive SLSs in either population.” It could be added here what kind of evidence would be required to prove the existence of explicitly metacognitive SLSs in animals, especially because above it was mentioned that these SLSs are reportable rules – if this means verbally reportable, then the existence of such SLSs would a priori not be possible in animals. This should be clarified in the manuscript.

Introduction, l. 110-115: “adult humans are (with good reason) assumed to be able to use social information in an explicitly metacognitive manner […] For example, it is routine for human adults to actively seek out models of social behaviour, using their understanding of others’ intentions relative to their own to strategically select and use the most relevant social information […]. A critical question, then, is how children’s use of social information develops with age, particularly in relation to their explicit understanding of the value of social information” The use of explicitly metacognitive SLSs is here exemplified by knowing which social models to seek out and using others’ intentions to select information. Then the question is asked how these abilities develop in children. However, the task used in the current study does not require a selection between social models or reading others’ intention – instead, as several comments of mine show, it does not really become clear in what way the task used here is social. It seems that children just need to focus on the boxes shown in the video stills (which box looks like the one I have in front of me?) and that the demonstrators can be ignored. The value of information lies in the type of boxes, not in the models themselves. In short, I find that there is a gap between the theoretical description of explicitly metacognitive SLSs and what they entail on the one hand and how this is implemented in the current task. This information should be added to allow the reader to follow the argument.

Introduction, l. 134-136: “in the current study we were particularly interested in determining when children develop the ability to seek out social information based on an understanding of the value of that information.” It should be explained why the inclusion of “social” is warranted here. It seems that the task used just tested the ability to seek out information based on an understand of the value of this information.

Introduction, l. 183-188: “While it is useful to know how children use social information and who children prefer to learn from, we argue that this is not sufficient to determine the cognitive mechanisms that children are employing during social learning. Children’s ability and proclivity to seek out relevant social information has not yet been adequately addressed. That is, nothing in the SLSs or selective trust literature has examined children’s ability to select valuable social information on the basis of its relevance for solving a specific problem.” Could this question be rephrased to “when do children become able to find the correct solution to a problem?” Again, as mentioned above, I am struggling to find the reasoning for including “social” here. Also, would the answer to the question not also depend on the specific problem and the experience that children have related to the problem? This should be discussed.

Introduction, l. 244-248: “Appropriate use of social information (following appropriate information seeking) could further help to distinguish between learners with some level of metacognitive understanding, and those reliant on implicit rules for what, when, and whom to copy. That is, understanding the relevance of the acquired information would be expected to also result in more appropriate use of that information.” I would suggest adding a bit more information on how exactly the use of social information can provide additional insights into the use of social learning strategies over and above the seeking out of relevant information. This becomes a bit clearer when reading the discussion of the paper, but is not very clear at the introduction. Adding an example to illustrate this point would be helpful here to let the reader follow the argument. In other words, how exactly can finding out how children use social information tell us about the nature of their social learning processes in addition to finding out how children seek the information?

Methods, 376-379: “Appropriate information seeking was measured by the proportion of trials in which children selected the target video. This required children to reason about the information they needed to solve a problem (unlock their box) and which of the demonstrators could provide that information (which of the demonstrators were facing the same problem).” The second sentence suggests that children would have to attend to two kinds of sources of information to make the correct choice: the box depicted in the video and the demonstrator. It is unclear to me why attention to the demonstrator is needed – children could just select the video that shows the same problem as theirs and disregard the demonstrator. Indeed, this is the point of the study: “If children were reasoning appropriately, they should have disregarded demonstrator characteristics” (l. 379/380). Therefore, this second sentence should be rephrased as appropriate information seeking did not require reasoning about which demonstrator could provide information.

Discussion, l. 527-530: “Specifically, we were interested in identifying when children develop the ability to seek out information based on a reasoned understanding of what information is required to solve a particular problem and who could provide it (explicitly metacognitive SLSs).” Similar to the previous comment, I am questioning whether being successful in the task used in the current study required both attending to what information is required and who is able to provide it. It seems that attending to what information is required was sufficient. If both information and demonstrator had to be attended to, the design should have been different, e.g., there being two or more demonstrators with the correct box, but only one of them being able to provide the information for whatever reason (e.g., if children wouldn’t be able to watch the video, but just saw the still, and then had to rely on a verbal report (in a different video) by the demonstrators, but for some reason only one demonstrator could provide that report (e.g., the others cannot speak or cannot be heard). I suggest rephrasing this sentence accordingly or to explain in more detail why attention to both the type of box and the demonstrator is required.

In addition, also referring to the same sentence, the discussion section fails to answer this question. While it is discussed that children’s performance in the task increases with age, no answer is given to when the ability to seek out information based on its value emerges. As performance in all age groups was above chance levels, I assume the answer here would be that this ability emerges very early in life, before 3 years of age. This should be added and discussed.

Results: I am wondering whether the authors have investigated whether there is a trial effect on whether or not mistakes are made, i.e., whether the unsuccessful trials tended to be the first trials (especially because there did not seem to be a warm-up or a check to see whether children had understood what to do). I am wondering whether errors occurred because children initially did not quite understand what to do and then fell back on an implicit SLS or there was no trial effect, indicating that some of the errors were made not because the children did not understand the task, but because a pull of an implicit SLS was too strong.

Discussion, l. 535/536: “We found that the proportion of trials in which children selected the target demonstration video increased with age. This corresponded to our prediction that older children would exhibit more appropriate social information seeking than younger children.” Again, more explanation is needed as to why the behaviour investigated in this study can be labelled social information seeking. What is the social here, especially because a correct response to the task would imply disregarding social information (i.e., information about demonstrator characteristics)?

Discussion, l. 544-552: “Thus, we suggest that older children understood the potential value to themselves of the information that the target demonstrator could provide having employed an explicitly metacognitive SLS. By contrast, we postulate that younger children’s target video selections were less likely to have been driven by reasoned understanding of the information they needed and the potential value of the target demonstration video. Therefore, when selecting a video, younger children might have instead relied on less cognitively demanding implicit SLSs such as model-based biases, or a bias for selecting models whose box matched their own box (without appreciating the value of the information).” More information needs to be provided (already in the introduction, and possible again here) why choosing the correct video requires an “explicitly metacognitive” understanding of the information that was needed. What about alternative explanations, e.g., that simply recognizing the box that is in front of oneself in one of the video captures? Second, I might be misunderstanding what children needed to do in the task (and if so, this should be made clearer) but did they not have to select the video with the model whose box matched their own box? I am struggling to follow the argument that young children might have had a bias for selecting models whose box matched their own. More information should be added here to allow the reader to follow the argument. In general, I think that the introduction could include more specific information on what specific demands the task poses and why.

Discussion, l. 563-565: “Results showed that children were more likely to select target demonstrators who were the same gender as themselves, therefore showing a gender congruence effect.” I am not convinced that the data support this conclusion. The gender congruence effect was only observed in girls, and so does not apply to the entire sample. Thus, the conclusion should be adjusted accordingly.

The following point as several comments: Discussion, l. 568-570: “Therefore, it seems that even when it was apparent – from salient visual cues – which demonstrator could provide relevant information”. Related to a previous comment, I find this phrasing misleading, as this suggests that solving the task needs attention to the demonstrator at all or asking the question “which demonstrator provides the information I need”. However, the visual cues that are mentioned here relate to the box only, not to the demonstrator. I suggest clarifying this and similar sentences throughout the manuscript.

Another issue is that the relevant information lies outside of the demonstrator (box) and not within them. That is, the task could also be solved if the videos only showed the boxes and them being opened with the demonstrator and their hands digitally removed. Therefore, also related to a previous comment, the task is not inherently social it seems to me, or at least it should be better explained why it should be regarded as one.

Adding the demonstrator just adds the potentially distracting information on age and gender. The study could also have used a different design, in which the relevant information is “inside” the demonstrator, e.g., via a knowledge state, and could contrast with heuristic biases. Why was it chosen to have the value of information in the separate object rather than keeping it attached to the demonstrator?

The fact that there are two potential sources of information (object and demonstrator) puts a (possibly for the current research question unnecessary) demand on attentional resources on children. Therefore, it could be possible that younger children were overwhelmed by the richness of the information provided in the videos and that this could have led them to fall back on simpler strategies rather than them being unable to seek out the correct information per se. This potential limitation should be discussed.

Discussion, l. 627-630: “However, as we used age as a proxy for cognitive development and developmental trajectories of other cognitive capacities overlap with that of explicit metacognition, we cannot say for certain which cognitive developments might be implicated in the emergence of the capacity for explicitly metacognitive SLSs.” It could be added here that future studies would need to investigate whether individual differences in the use of explicitly metacognitive SLSs correlate with differences on other tasks measuring these other putative cognitive developments.

Discussion, l. 653: The discussion mentions that children in the current study were able to actively seek out appropriate social information. As mentioned above, an explanation is needed why the information provided in the videos is social.

Information should be provided on the question whether it can be ruled out that the younger children failed to understand the task, e.g., that they could only choose one key, and also whether the younger children were motivated enough (I understand they could not keep the rubber duck?). Did the younger children understand at all that they were in need of more information?

Discussion, l. 667-671: “explicitly metacognitive SLSs are experience-dependent and develop relatively late in children. The relatively late emergence of the capacity to use reasoning-based, or explicitly metacognitive, SLSs suggests that this may be a cognitive mechanism that is unique to humans and is unlikely to be observed in animals”. The argument hat SLSs are experience-dependent and that they may rely on uniquely human cognitive abilities are two different, but not mutually exclusive, possibilities. This should be made a bit clearer: what is it that younger children are likely to lack? Certain cognitive mechanisms or general experience from the social and physical world?

Minor points:

- Introduction, l. 40-43: “Seeking out relevant information from appropriate social sources is ubiquitous in human adults. Human adults may therefore demonstrate key differences in the way they seek, attend to, and use social information compared to children and non-human animals (henceforth animals).” I would suggest deleting “therefore”, as the suggestion described in the second sentence does not follow logically from the first sentence.

- Introduction, l. 156: “Similar to the selective preferences in children’s proclivity to copy and consistent” – please doublecheck this sentence, it seems as if “consistent” should be a verb or if a verb is missing

- Methods, l. 318: “experimenter one” – change to “experimenter 1”

- Methods, l. 358: “cantered” – change to “centered”

- Results, l. 434: “demonstrators” children” – remove apostrophe?

6. PLOS authors have the option to publish the peer review history of their article (what does this mean?). If published, this will include your full peer review and any attached files.

Reviewer #1: **Yes: **Ángel V. Jiménez

Reviewer #2: No

---

## [Author Response · Author response to Decision Letter 0]

30 Jul 2021

Responses to reviewer comments have been given in the uploaded 'Response to Reviewers' file.

---

## [Editor Report · Decision Letter 1]

11 Aug 2021

Development of strategic social information seeking: Implications for cumulative culture

PONE-D-21-09287R1

Dear Dr. Blakey,

Thank you for taking the time to revise your paper. I am satisfied with your responses to the reviewers' feedback and the associated changes in the manuscript, and am therefore pleased to inform you that your manuscript has been judged scientifically suitable for publication and will be formally accepted for publication once it meets all outstanding technical requirements.

Kind regards,

Mark Nielsen, Ph.D.

Academic Editor

PLOS ONE

---

## [Editor Report · Acceptance letter]

16 Aug 2021

PONE-D-21-09287R1 

Development of strategic social information seeking: Implications for cumulative culture 

Dear Dr. Blakey:

I'm pleased to inform you that your manuscript has been deemed suitable for publication in PLOS ONE. Congratulations! Your manuscript is now with our production department. 

Kind regards, 

on behalf of

Dr. Mark Nielsen 

Academic Editor

PLOS ONE